# The Efficacy of an Ultrasound-Guided Improved Puncture Path Technique of Nerve Block/Pulsed Radiofrequency for Pudendal Neuralgia: A Retrospective Study

**DOI:** 10.3390/brainsci12040510

**Published:** 2022-04-18

**Authors:** Dan Zhu, Zhenzhen Fan, Fujun Cheng, Yuping Li, Xingyue Huo, Jian Cui

**Affiliations:** 1Department of Pain Medicine, Southwest Hospital, Army Medical University, Chongqing 400038, China; zd13752902138@163.com (D.Z.); zhenzhenf1101@163.com (Z.F.); hbyyxymz@163.com (F.C.); lyp536_mzk@163.com (Y.L.); 2Department of Population Science and Health Policy, Icahn School of Medicine at Mount Sinai, New York, NY 10029, USA

**Keywords:** pudendal neuralgia, pudendal nerve block, pulsed radiofrequency, improved puncture path, ultrasound guidance

## Abstract

Objectives: To investigate the efficacy and safety of an improved ultrasound-guided pulsed radiofrequency (PRF) and nerve block (NB) for patients with pudendal neuralgia (PN). Methods: This retrospective analysis included 88 adults with PN treated in the Pain Department of Southwest Hospital from November 2011 to June 2021, with treatment including NB (*n* = 40) and PRF (*n* = 48). The primary outcome variable was pain severity, measured by a standardized visual analog scale (VAS). VAS values were collected at 1, 3, 7, and 14 days and 1 and 3 months after patients were treated with NB or PRF. Results: Compared with patients treated with NB (*n* = 40) and those treated with PRF (*n* = 48), no significant difference in pain reduction was observed in the short term (*p* = 0.739 and 0.981, at 1 and 3 days, respectively); however, in the medium and long term (1 to 3 months), there were statistically significant improvements in the PRF group over the NB group (*p* < 0.001). Moreover, it was noted that the average pain severity of primary PN and PN due to sacral perineurial cyst was significantly reduced with PRF therapy in the medium and long term when compared to other secondary PNs, including surgery, trauma, and diabetes. Discussion: The ultrasound-guided, improved, and innovative PRF/NB puncture path technique allows for gentler stimulation and faster identification of the pudendal nerve. The PRF technique may provide better treatments for primary PN and sacral perineurial cyst causing secondary PN in the medium and long term.

## 1. Introduction

Pudendal neuralgia (PN) is a well-known medical challenge involving the motor (20%), sensory (50%), and autonomic (30%) functions of the pudendal nerve, with a prevalence of approximately 6.6% in the population in general [1,2]. The early treatment and management of this neuropathic pain are still challenging tasks [3,4]. Treatments including analgesics or neuroactive drugs, pudendal nerve block, and surgical decompression therapies are ineffective for over 50% of patients [5,6,7].

Accurate pulsed radiofrequency (PRF) therapy combined with nerve block is a novel method to improve the function of pudendal nerve and to relieve the pain of PN patients [8,9]. In recent years, ultrasound-guided pudendal nerve block and/or PRF techniques have been widely used due to the accuracy, visualization, and lack of radiation [6,7,10]. Compared to continuous radiofrequency (CRF), PRF is a safer and nondestructive technique [11,12]. To assure the efficacy of treatment for patients with neuropathic and chronic pain, the precise diagnosis of PN and the identification of the pudendal nerve and adjacent anatomical structures are of great importance. An equally significant aspect of this procedure is the skill of puncturing the path under ultrasonic real-time guidance to ensure that the needle tip can approach as close to the pudendal nerve as possible without injuring of essential vessels [5,13].

At present, only a few cases and clinical trials involving nerve block or PRF techniques for refractory PN have been conducted due to insufficient study populations and follow-ups. The objectives of this retrospective study are as follows: (1) to find a novel, simpler, and faster puncture path for ultrasound-guided stimulation and confirmation of the pudendal nerve; (2) to compare the therapeutic efficacy and safety between PRF and NB therapies in the medium to long term; and (3) to further discuss and analyze the efficacy of ultrasound-guided PRF for the therapy of secondary and primary PN.

## 2. Patients and Methods

### 2.1. Data Sources

This retrospective study was approved by the ethics committee of the First Affiliated Hospital of the Army Medical University, PLA (permit number: KY2020172, Chongqing, China). Hospitalized patients who visited the pain department at Southwest Hospital from November 2011 to June 2021 (Figure 1) were eligible for inclusion in the trial after being diagnosed with the Essential Diagnostic Nantes Criteria [14] as well as the following inclusion protocol: (1) The range of pain occurs in the area of the pudendal nerve; (2) the pain is mainly obvious in the sitting position; (3) the patient does not wake up at night due to pain; (4) there is no objective sensory disturbance of the local pain area during physical examination; (5) the pain is relieved by diagnostic pudendal nerve block; (6) patient who completed imaging and laboratory examination, such as abdominal computed tomography (CT), lumbosacral iliac joint magnetic resonance (MRI), and blood routine; and (7) the patient agreed to treatment with a pudendal nerve block (NB) or pulsed radiofrequency (PRF). The exclusion criteria were as follows: (1) tumors of the pelvic or/and sacral channel were found with abdominal computed tomography (CT), lumbar, and sacroiliac joint magnetic resonance imaging (MRI), and other examinations; and (2) symptoms only presented with tail bone, hip, or hypogastric pain and itching.

### 2.2. Data Collocation

We included age, gender, VAS, and short-term (1, 3, 7, and 14 days post-treatment) and med-/long-term (1 to 3 months post-treatment) results as variables for the patients treated with NB and/or PRF. Pain severity and pain reduction were reassessed and standardized from the patient’s initial VAS score level to the latest recent VAS score level to increase the objectivity of the data. Lower VAS scores indicated less severe pain, and higher VAS scores referred to more severe pain. Data on pain severity were collected by telephone or Wechat video call follow-up.

### 2.3. Conventional Pudendal NB and Ultrasound-Guided Improved Pudendal NB

#### 2.3.1. Conventional Pudendal NB Technique

For patients with PN in the lithotomy position, a previous study described the conventional pudendal NB approach taken [15]. In brief, the physician uses a finger to locate the patient’s ischial spine along the vaginal/rectal wall, and then a 22 G × 0.7 mm × 80 mm needle is inserted into the interior of the ischial spine. After the syringe is pulled back blood-free, a 5 mL volume is injected, including a mixture of 1.5 mL of 2% lidocaine, 2.5 mL of 1% ropivacaine, and 5 mg of triamcinolone.

#### 2.3.2. Ultrasound-Guided Improved Pudendal NB Puncture Path Technique

After positioning the PN patient in a prone position (Figure 2), a low-frequency ultrasound curved array transducer (SIEMENS, KT-LM170SDS, ACUSON X300, Malvern, PA, USA) is placed in the area of the ilium (Figure 3A), which corresponds to the anatomical position (Figure 3B) (VESAL 3D ANATOMY, Xi’an, China, 4.5.2). After identifying the continuous hyperechoic line of the ilium area, the ultrasound transducer is slowly moved parallel downward from superior to inferior to reveal the three key anatomical position areas, including the piriformis, ischial spine, and underneath the ischial spine between the ischial spine and ischial tuberosity (Figure 3C–H), and then moved slightly back to the plane of the ischial spine. The physician localizes the pudendal nerve around the internal pudendal artery at the plane of the ischial spine under ultrasound direction and then uses a 22 G × 0.7 mm × 80 mm needle to puncture the internal pudendal artery laterally and then injects 5 mL of the mixed liquid (the same concentration and dose of drug as the conventional pudendal NB group).

#### 2.3.3. Ultrasound-Guided Improved Pudendal NB and PRF Puncture Path Technique Group

The day before a patient with PN receives PRF treatment, physicians must determine the effectiveness of pudendal NB therapy under ultrasound guidance. We administer PRF therapy only when the pain severity in pudendal NB is reduced by more than 50% within 2 continuous hours by pudendal NB therapy. An RF needle (22 G × 100 mm, 240102, Emmendingen, Germany) with a 5 mm active tip is placed laterally to the internal pudendal artery and is observed to generate a sensory abnormality by stimulation testing (Bei Qi, China, R-20000BM2) at a frequency of 50 Hz and a voltage of 0.1–0.5 V. The parameters of PRF ablation are then delivered at a frequency of 2 Hz, a pulse width of 20 msec, and a voltage of 45 V and 42 °C for 240 s, with the procedure conducted in two cycles. A total of 5 mL of the compound liquid with 2% lidocaine 1.5 mL, 1% ropivacaine 2.5 mL, and triamcinolone 5 mg is administered after the PRF application. The same PRF procedure is repeated after 5–7 days, with 2 PRF treatments during hospitalization.

### 2.4. Statistical Analysis

In this study, the primary outcome was VAS, a standard measure of pain severity. The pain severity ratio was used as a VAS relative ratio, which measured the VAS score at a given time point (post treatment score) over the baseline (pretreatment score) for each patient. A chi-square test was used to demonstrate whether a significant relationship existed between treatment groups and the characteristic variables, including sex, side, and etiology. Thus, a significant result meant that there was a significant difference in pain severity in the characteristic variables. A t-test was applied to test for equality of the variables of age and pain severity within the NB/PRF group. Post treatment VAS values and pain severity at different post treatment time points (1 d, 3 d, 7 d, 14 d, 1 M, 3 M) were compared with baseline (pretreatment), respectively, with paired t-tests to analyze whether there were significant changes in pain severity pre/post treatment. One-way ANOVA tests were conducted for 4 types of secondary PN comparison. Statistical software SPSS 19.0 (SPSS Inc., 2009. IBM Chicago, IL, USA) was used to analyze the VAS values, and pain severity of the independent variable groups was defined as significantly different at the 0.05 level.

## 3. Results

### 3.1. Subject Characteristics

Phone follow-up of the 88 subjects was conducted at 1, 3, 7, and 14 days, and 1 and 3 months after they completed NB or PRF treatment. The follow-up was terminated if there was no treatment effect. No significant side effects were observed in any treated patients. The clinical characteristics of the PN patients are listed in Table 1. The NB and PRF groups included 40 subjects with an average age of 51.9 (SD = 13.0) and 48 subjects with an average age of 59.4 (SD = 14.1), respectively. The clinical characteristics, including sex, age, side, and etiology, were not significantly different between the NB and PRF groups (*p* > 0.05, NB vs. PRF, Table 1).

### 3.2. Comparison between Conventional and Improved Pudendal NB

The NB group consisted of 40 patients, including 22 with conventional pudendal NB and 18 with ultrasound-guided improved pudendal NB puncture. There was no significant difference in pain severity from D1 to M3 after NB treatment (*p* > 0.05, conventional vs. improved, Table 2). Furthermore, the standardized VAS value for pain severity was obviously decreased from D1 to D3 posttreatment compared to baseline in the conventional/improved pudendal NB group, suggesting a positive treatment result (*p* < 0.001, BS vs. 1 and 3 days, Table 2). However, there was no significant change in efficacy from D7 to M3 compared to baseline with conventional or improved pudendal NB treatment (*p* > 0.05, BS vs. 7, 14 days, 1 and 3 months, Table 2).

### 3.3. Comparison between NB and PRF

Based on the results shown in Table 2, both conventional and improved pudendal NB treatments could effectively alleviate neuralgia in the short term (within 3 days after treatment) but not in the medium or long term. Therefore, the efficacy of PRF treatment of PN patients should be observed over a longer period of time. As shown in Table 3, there was no significant difference in the efficacy of treatment between the NB and PRF groups from pretreatment to D3 (*p* > 0.05, NB vs. PRF, Table 3). However, compared with the NB group, the standardized VAS values for pain severity decreased significantly from D7 to M3 in the PRF group (*p* < 0.001, NB vs. PRF, Table 3). Moreover, the effect of PRF treatment for PN gradually declined over time compared to the NB group, with a change in pain severity from 26% to 59.8% (mean (%), Table 3) between D1 and M3. In summary, PRF treatment significantly improved the treatment outcomes for patients with PN in the medium and long term compared to patients treated with NB.

### 3.4. Comparison between Secondary and Primary Pudendal Neuralgia Treated with PRF

We further analyzed whether the etiology of PN, including secondary (*n* = 28) and primary (*n* = 20) PN, had an impact on the outcome of PRF treatment. As shown in Table 4, there was no significant difference in pain severity values between secondary and primary PN within 14 days after PRF treatment (*p* > 0.05, secondary vs. primary, Table 4). There was, however, a significantly improved result of PRF treatment for primary PN compared to patients with secondary PN at 1 month (*p* < 0.05, secondary PN vs. primary PN, Table 4) and 3 months (*p* < 0.001, Table 4). These results suggested that the PRF treatment of primary PN had better efficacy than for secondary PN, especially in the medium and long term (1 month and 3 months).

### 3.5. Efficacy for Various Etiologies of Secondary Pudendal Neuralgia with PRF

The etiology of secondary PN patients who had received PRF treatment in our Pain Medicine department included surgery (*n* = 6), trauma (*n* = 12, occurring after urinary tract infection or sexual activity), sacral perineurial cyst (*n* = 6), and diabetes (*n* = 4). As shown in Table 5, the pain severity values of secondary PN due to a sacral perineurial cyst (*n* = 6) slowly and gradually increased from 21.7% to 43.3% compared to 22 other secondary PN cases in the medium and long term (1 month and 3 months, mean (%), Table 5), including those due to surgery, trauma, and diabetes. This increase in pain severity means a decrease in efficacy. Another finding was that the values of pain severity were similar for the other three types of secondary PN and were not significantly reduced after treatment over the medium and long term (1 month and 3 months, mean (%), Table 5). It is worth noting that the pain severity values of sacral perineurial cyst was significantly reduced compared with the other three types of secondary PN at 1 month after treatment (*p* < 0.05). All of these results suggested that although some secondary PN is not significantly relieved in the medium and long term after PRF treatment, it could be possible that PRF treatment of secondary PN patents with sacral perineurial cyst results in improvements in the medium and long term.

### 3.6. Degree of Pain Relief (>50%) in Pudendal Neuralgia

To continue investigating the degree of pain relief in patients with pudendal neuralgia from baselines (pretreatment) to post treatment (1d, 3d, 7d, 14d, 1M, 3M), we set relief values > 50% as treated effectively [16,17] (Table 6). A standardized VAS was used to analyze and measure the pain relief ratio for each patient at a certain time point. For the baseline (pretreatment) values, the cases of pain relief were set to 0%. Except for primary PN and sacral perineurial cyst from secondary PN, pain reduction of more than 50% decreased and was maintained at a constantly lower level as early as 1 day after PRF/NB treatment to 3 months. As shown in Table 6, the effective pain reduction of NB and PRF was ascertained in the short term (1 and 3 days) and medium and long term (1 to 3 months), respectively. In addition, the effective pain reduction of secondary PN was significantly lower than primary PN in the medium and long term. Particularly, the percentage of secondary PN including surgery, trauma, and diabetes was decreased from 100% to 16.7%, from 100% to 16.7%, and from 100% to 0% in pain reduction, respectively. According to the above findings, PRF in medium- and long-term efficacy in patients with PN is commendable. Pulsed radiofrequency has a precise medium- and long-term efficacy in patients with pudendal neuralgia, especially in patients with primary PN and sacral perineurial cyst from secondary PN.

## 4. Discussion

In the present study, we explored a novel and improved puncture path technique to alleviate refractory pain in the medium and long term by observing the therapeutic outcomes of patients with PN who were treated with NB (*n* = 40) or PRF (*n* = 48). Moreover, the new and improved puncture path under ultrasound guidance was evaluated for the first time. The therapeutic efficacy of PRF for the treatment of patients with secondary and primary PN was then further analyzed.

The standardized pain severity measured by the VAS was significantly decreased by 70% or more in both the NB and PRF groups within 3 days (short term) after treatment. In patients with PN who were considered to be refractory to oral systemic lamotrigine and opioids, PRF treatment significantly relieved pain, especially in the medium and long term (1 month and 3 months). Furthermore, we observed that the pain reduction in the medium and long term (1 month and 3 months) was higher in patients with primary PN treated with PRF than in patients with secondary neuralgia. Another important finding was that PRF treatment of secondary PN with a sacral perineurial cyst could reduce the severity of pain compared to the other three etiologies of secondary PN in the medium and long term.

The observation data are from 2011 to 2021, which is a long time span. However, we did follow up the treatment effect by telephone or Wechat video call, and from 2011 to 2015, our treatment for pudendal neuralgia was nonvisualized pudendal nerve block, and only five patients were lost to follow-up. We started using ultrasound-guided pudendal nerve block and pulsed radiofrequency therapy in 2016. Therefore, the long time span has little effect on our ultrasound-guided nerve block and pulsed radiofrequency results.

At the ischial spine level, the average diameter of the pudendal nerve is only approximately 5 mm [8]. Despite being visually ultrasound-guided, knowledge of the pudendal nerve and the complex adjacent anatomical structure is essential to the completion of NB or PRF therapy. The pudendal nerve originates from S2–4, passes through the greater sciatic foramen, and crosses as deep as the piriformis, where it crosses between the sacrospinous and sacrotuberous ligaments and is accompanied by the internal pudendal artery at the level of the ischial spine [2,18]. In addition, there are three terminal branches of the pudendal nerve, including the dorsal nerve of the inferior anal nerve, perineal nerve, and penis/clitoris nerve [2,19], which spread across the skin of the rectum, anus, scrotum/labia, and penis/clitoris, respectively. Therefore, whenever a stimulus can successfully elicit a sensation in one of these three areas, it indicates the discovery of a pudendal nerve.

In the current study, under ultrasound guidance, the improved puncture approach is from the lateral side of the transducer. In a previous study, the puncture needle was considered an in-plane approach to the conventional pathway from the medial transducer [10,20]. However, possible limitations of the puncture path from the medial transducer are as follows: (1) As the puncture path in that manuscript shows [10], the angle of the puncture needle was too vertical (about 70°) to enable maximal access to the region between the 5 mm active tip and the pudendal nerve [10]; (2) in the area of the ischial spine, the pudendal nerve crosses the internal pudendal artery from lateral to medial [18], and inducing sensory stimulation is difficult if the pudendal artery is located inside the pudendal nerve and the puncture angle is too vertical; and (3) due to the anatomical variation of the pudendal nerve at the ischial spine level [21], the pudendal nerve or branches may be closer to the sciatic and inferior gluteal nerves [22,23,24]. The active tip of the PRF needle may induce other sensory/motor stimulation that is nonpudendal-related. Therefore, the ultrasound-guided improved puncture path, as the puncture needle moves toward the medial side (Figure 2), can avoid the sensory/motor stimulation of the sciatic and inferior gluteal nerves caused by tip contact, thereby alleviating any discomfort and increasing the area of the active tip contacting the pudendal nerve.

All of the patients with PN in this analysis received a 5 mL liquid mixture of lidocaine, ropivacaine, and triamcinolone. In this retrospective study, patients in both the conventional and improved NB treatment had an onset of pain relief approximately 3 days after treatment. It was assumed that the effectiveness of local anesthesia is predominant in the short term, although the half-life of triamcinolone is close to 30 days, which is consistent with other studies [25,26]. During medium- and long-term follow-up, PRF treatment alleviated pain severity in patients with PN more effectively than NB.

The most common sequence is to perform PRF with the parameters of 2 Hz pulse frequency, 45 V, 20 msec pulse width, and 42 °C [10,27]. However, the relationship between multiple repetitions of PRF, a high pressure or long-term operation, and the therapeutic effect still needs to be observed and confirmed by a large number of subjects and extensive experience with short- and long-term follow-up. The safety profile of PRF is evident and indisputable, although the therapeutic mechanism of PRF is not known [11]. In the current study, no adverse events were reported in patients treated with PRF therapy.

From a long-term perspective, patients with primary PN receiving PRF treatment had better outcomes than those with secondary PN, from which it is speculated that secondary factors of surgery and trauma may lead to adhesions between the local muscle fascia and the pudendal nerve or branches. In PRF treatment, moreover, secondary PN with diabetes also had poor therapeutic results compared to primary PN in the medium and long term, so patients may need further strict control of their blood glucose fluctuations in the future. Another key finding is that PRF treatment had better efficacy for treating PN caused by sacral perineurial cysts than PN caused by the other three secondary factors, especially during the medium and long term. Sacral perineural cyst, also known as Tarlov cyst, is a cystic nerve root lesion that is common in the sacrum [28,29]. Elsawaf et al. reported that patients had pain symptoms, mostly with the smallest size being 2 cm [28]. In this study, however, only six PN patients with sacral perineurial cysts, with cysts less than 1 cm in size, were observed. As a result, it is unclear yet if the pudendal neuralgia is caused by the sacral perineurial cyst, or if it should be classified as primary pudendal neuralgia. Additional studies and more subjects are required to confirm the efficacy of PRF treatment for sacral perineurial cyst patients in future clinical trials.

Finally, we used the VAS to measure pain intensity; a higher VAS score indicates greater pain intensity. Nevertheless, pain reporting may vary according to subjective factors; the same intensity of pain may be reported differently by different individuals. Thus, the pain severity ratio was used as a VAS relative ratio, which measured the VAS score at a given time point (post treatment score) over the baseline (pretreatment score) for each patient. Furthermore, the results indicated a high standard deviation in pain severity after PRF treatment in patients with PN, especially in those with secondary PN. These patients with secondary neuralgia, including surgery, trauma, sacral perineurial cyst, and diabetes, were administered PRF treatment, and only patients with a sacral perineurial cyst had a significant average reduction in pain severity during the medium and long term, which may be the reason for the large standard deviation.

The study’s limitations included a single-center design and a small number of patients. Auxiliary technologies, such as virtual reality (VR) and 3D simulation would be required in the future to improve research precision and practicality.

## 5. Conclusions

In this study, ultrasound-guided, improved, and innovative pudendal NB and PRF puncture path techniques were proven to be safe and effective for PN patients and were more easily implemented once we understood the pudendal nerve and its adjacent anatomical structure. In addition, the PRF technique showed significant improvement in the treatment of primary PN and PN caused by sacral perineurial cyst in the medium and long term. The results of the current study, as well as the ultrasound-guided improved path technique, are particularly important and meaningful for the treatment of PN and should be widely applied.

## Figures and Tables

**Figure 1 brainsci-12-00510-f001:**
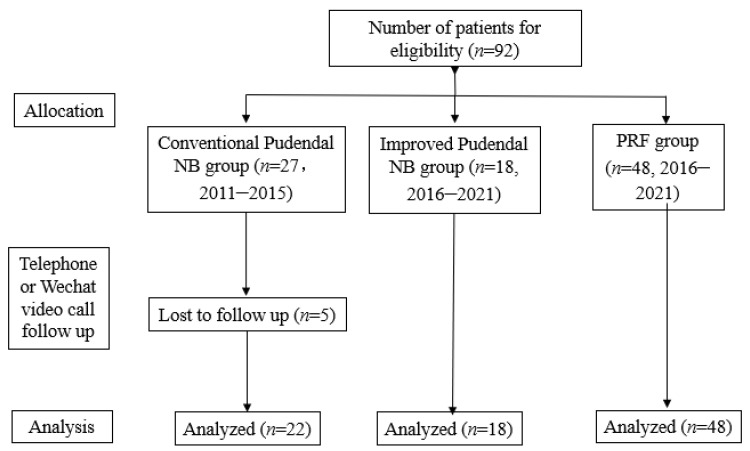
The flow chart is shown.

**Figure 2 brainsci-12-00510-f002:**
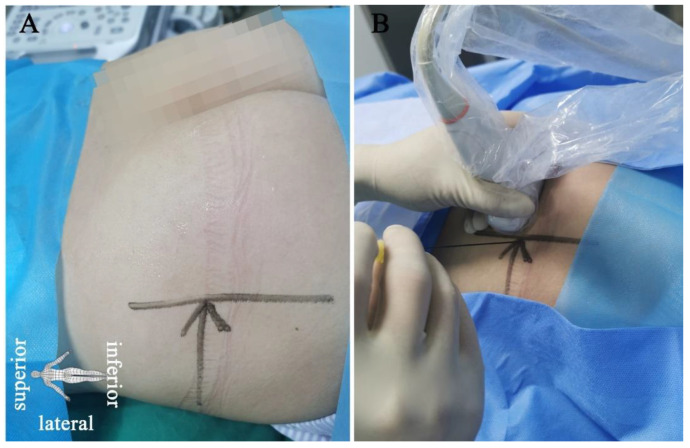
Positioning of the patient and the puncture with the ultrasound-guided improved pudendal NB/PRF technique. Patient with pudendal neuralgia in the prone position (**A**) and needle punctures in-plane into the lateral end of the low-frequency ultrasound transducer (**B**).

**Figure 3 brainsci-12-00510-f003:**
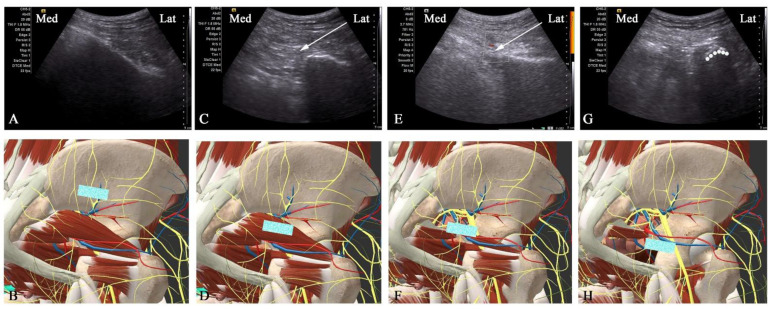
The key anatomical locations for ultrasonic transducer placement (blue identification in (**B**,**D**,**F**,**H**)) and the corresponding ultrasound images (**A**,**C**,**E**,**G**). (**A**,**B**) The ultrasonic transducer is placed in the ilium and showed continuous hyperechoic lines. (**C**,**D**) The transducer is moved downward parallel to the greater sciatic foramen and then reveals the piriformis (white arrow marks). (**E**,**F**) The transducer continues to move parallel to the ischial spine so that the internal pudendal artery, a branch of the internal iliac artery adjacent to the pudendal nerve, could be seen on the color Doppler. The inserted needle (white arrow), lying laterally to the internal artery, visualizes the puncture path and the target pudendal nerve. (**G**,**H**) The transducer is slowly moved parallel to the hypoechoic crescent (dashed line), which shows the area underneath the ischial spine between the ischial spine and ischial tuberosity. This helps us to identify the position of the ischial spine and pudendal nerve.

**Table 1 brainsci-12-00510-t001:** Demographic characteristics of all patients. NB = nerve block, PRF = pulsed radiofrequency.

		NB (*n* = 40)	PRF (*n* = 48)	*p* Value
Gender (%)	Female	30 (75%)	32 (66.7%)	0.546
	male	10 (25%)	16 (33.3%)	-
Age (Mean ± SD)		51.9 ± 13.0	59.4 ± 14.1	0.073
Side (%)	Unilateral	16 (40%)	14 (29.2%)	0.450
	bilateral	24 (60%)	34 (70.8%)	-
Etiology (%)	secondary	18 (45%)	28 (58.3%)	0.316
	primary	22 (55%)	20 (41.7%)	-

**Table 2 brainsci-12-00510-t002:** Changes in VAS and pain severity between the conventional and improved pudendal NB groups and changes in VAS and pain severity from baseline in the conventional/improved pudendal NB group. NB = nerve block, VAS = Visual Analog Scale, BS = baseline, D = day, M = month.

	Conventional Pudendal NB (*n* = 22)					Improved Pudendal NB (*n* = 18)					*p* Value (Conventional vs. Improved)
	VAS		Pain Severity		*p* Value (BS vs. Point-In-Time)	VAS		Pain Severity		*p* Value (BS vs. Point-In-Time)	
	Mean	SD	Mean (%)	SD		Mean	SD	Mean(%)	SD		
baseline	6.6	0.7	100%	0.11	-	6.6	0.83	100%	0.12	-	1.0
1d	1.7	0.5	25.9%	0.07	*p* < 0.001	1.6	0.21	24.0%	0.05	*p* < 0.001	0.739
3d	1.9	0.6	29.6%	0.08	*p* < 0.001	1.7	0.35	29.0%	0.07	*p* < 0.001	0.977
7d	6.0	1.0	90.9%	0.13	0.091	5.8	1.25	87.8%	0.11	*p* < 0.05	0.573
14d	6.1	1.1	91.8%	0.12	0.105	6.1	1.27	91.1%	0.12	0.140	0.894
1M	6.2	1.1	92.7%	0.10	0.120	6.1	1.18	92.2%	0.10	0.162	0.911
3M	6.4	0.9	95.5%	0.07	0.255	6.2	1.03	94.4%	0.07	0.269	0.754

**Table 3 brainsci-12-00510-t003:** Changes in VAS and pain severity between the NB and PRF groups. NB = nerve block, PRF = pulsed radiofrequency, VAS = Visual Analog Scale, D = day, M = month.

	NB (*n* = 40)				PRF (*n* = 48)				*p* Value (NB vs. PRF)
	VAS		Pain Severity		VAS		Pain Severity		
	Mean	SD	Mean (%)	SD	Mean	SD	Mean (%)	SD	
baseline	6.61	0.75	100%	0.11	6.8	0.68	100%	0.10	1.0
1d	1.63	0.42	25%	0.06	1.8	0.32	26%	0.05	0.739
3d	1.84	0.51	29.5%	0.07	2.01	0.90	29.6%	0.14	0.981
7d	5.95	1.21	89.5%	0.12	1.97	0.87	31.0%	0.14	*p* < 0.001
14d	6.08	1.17	91.5%	0.11	2.23	0.91	32.3%	0.13	*p* < 0.001
1M	6.18	1.09	92.5%	0.10	2.71	1.48	40.8%	0.22	*p* < 0.001
3M	6.31	0.92	95.0%	0.07	4.01	2.30	59.8%	0.35	*p* < 0.001

**Table 4 brainsci-12-00510-t004:** Changes in VAS and pain severity between the secondary and primary pudendal neuralgia in the PRF treatment group. PRF = pulsed radiofrequency, VAS = Visual Analog Scale, D = day, M = month.

Improved Pudendal PRF									
	Secondary Pudendal Neuralgia (*n* = 28)				Primary Pudendal Neuralgia (*n* = 20)				*p* Value (Secondary vs. Primary)
	VAS		Pain Severity		VAS		Pain Severity		
	Mean	SD	Mean (%)	SD	Mean	SD	Mean (%)	SD	
baseline	6.64	0.68	100%	0.10	7.1	0.63	100%	0.93	0.153
1d	2.8	0.9	27%	0.04	1.71	0.46	23%	0.05	0.059
3d	2.19	1.05	31.8%	0.16	1.76	0.61	26.5%	0.10	0.367
7d	1.93	1.1	32.5%	0.17	2.03	0.43	29.0%	0.08	0.546
14d	2.4	1.1	35.0%	0.15	1.98	0.45	28.5%	0.09	0.236
1M	3.25	1.7	49.6%	0.25	1.96	0.47	28.5%	0.07	*p* < 0.05
3M	4.99	2.2	75.4%	0.33	2.65	1.70	38.0%	0.27	*p* < 0.01

**Table 5 brainsci-12-00510-t005:** Changes in VAS and pain severity from baseline in the secondary pudendal neuralgia group. * One-way ANOVA test for 4 types of secondary PN comparison, *p* < 0.05. PRF = pulsed radiofrequency, VAS = Visual Analog Scale, D = day, M = month.

Secondary Pudendal Neuralgia of PRF (*n* = 28)																
	Surgery (*n* = 6)				Trauma (*n* = 12, Occured After Urinary Tract Infection, after Sexual Life)				Sacral Perineurial Cyst (*n* = 6)				Diabetes (*n* = 4)			
	VAS		Pain Severity		VAS		Pain Severity		VAS		Pain Severity		VAS		Pain Severity	
	mean	SD	Mean (%)	SD	mean	SD	Mean (%)	SD	mean	SD	Mean (%)	SD	mean	SD	Mean (%)	SD
baseline	6.53	0.58	96%	0.08	6.4	0.84	94.1%	0.12	6.73	0.40	99%	0.06	7.35	0.21	108%	0.03
1d	2.4	0.17	30.0%	0	2.95	1.11	28.3%	0.04	2.5	1.06	26.7%	0.15	3.4	0.85	25.0%	0.07
3d	1.67	0.23	26.7%	0.06	2.32	1.31	35.8%	0.20	1.87	0.64	23.3%	0.15	3.1	1.27	40.0%	0.14
7d	2.63	1.44	41.7%	0.25	2	1.02	32.5%	0.17	0.9	0.35	21.7%	0.13	2.2	1.13	35.0%	0.07
14d	3.3	0.92	50.0%	0.15	2.65	1.01	37.5%	0.13	1	0.52	18.3%	0.07	2.5	0.42	30.0%	0.05
1M	3.47	1.96	56.7%	0.3	3.51	1.70	55.0%	0.23	1.47	0.80	21.7% *	0.13	4.75	0..64	65.0%	0.07
3M	5.4	2.31	81.7%	0.32	5.4	1.55	77.9%	0.23	2.87	3.17	43.3%	0.49	6.25	1.77	85.0%	0.21

**Table 6 brainsci-12-00510-t006:** Numerical statistics in degree of pain relief (>50%) from baselines (pretreatment) to post treatment (1d, 3d, 7d, 14d, 1M, 3M) in pudendal neuralgia.

	Degree of Pain Relief > 50% (*n*, %)							
	Conventional Pudendal NB (*n* = 22)	Improved Pudendal NB (*n* = 18)	Improved Pudendal PRF(*n* = 48)					
			Primary Pudendal Neuralgia (*n* = 20)	Secondary Pudendal Neuralgia (*n* = 28)				
				Surgery (*n* = 6)	Trauma (*n* = 12)	Sacral perineurial cyst (*n* = 6)	Diabetes (*n* = 4)	Total (*n* = 28)
baseline	0 (0%)	0 (0%)	0 (0%)	0 (0%)	0 (0%)	0 (0%)	0 (0%)	0 (0%)
1d	22 (100%)	18 (100%)	20 (100%)	6 (100%)	12 (100%)	6 (100%)	4 (100%)	28 (100%)
3d	22 (100%)	18 (100%)	20 (100%)	6 (100%)	11 (91.7%)	6 (100%)	4 (100%)	26 (92.8%)
7d	0 (0%)	0 (0%)	20 (100%)	4 (67%)	11 (91.7%)	6 (100%)	4 (100%)	24 (85.7%)
14d	0 (0%)	0 (0%)	20 (100%)	4 (67%)	10 (83.3%	6 (100%)	4 (100%)	24 (85.7%)
1M	0 (0%)	0 (0%)	20 (100%)	0 (0%)	6 (50%)	6 (100%)	0 (0%)	14 (50%)
3M	0 (0%)	0 (0%)	17 (85%)	1 (16.7%)	2 (16.7%)	6 (100%)	0 (0%)	8 (28.6%)

Complications: No complications occurred in all patients.

## Data Availability

The data presented in this study are available on request from the corresponding author. The data are not publicly available due to privacy or ethic.

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
