# Peer review of "The Efficacy of an Ultrasound-Guided Improved Puncture Path Technique of Nerve Block/Pulsed Radiofrequency for Pudendal Neuralgia: A Retrospective Study"

_brainsci, 2022, doi:10.3390/brainsci12040510_

Round 1

Reviewer 1 Report

This article focuses on interventional techniques for the treatment of pudendal neuroalgia. While the topic is of interest, I believe the article in its present form has some major issues. First, in the abstract, what has been identified as an objective is actually a background and the aim of the study is missing.

In the introduction, the gap of knowledge and aims are not quite well related and some phrases are confusing.  

One of the most relevant issues is the design of the study. While the authors state that this is a retrospective study, from the methods decription it seems like a prospective one with various follow-ups. Additionally, in the data collection, authors state they used the Visual Aanalogue Scales (VAS) scale to evaluate pain intensity, collected telephonically, which to my knowledge is not feasibile. 

In the statistical analysis too, it is unclear which timepoint was used as post-treatment score for the primary outcome. It is also not clear how were data of patients who early terminated follow-up treated.

Author Response

Apr. 6, 2022

Dear editor,

We are very grateful to Editors and Reviewers for reviewing the paper so carefully. We have carefully considered the suggestions of yours and make some changes. I am writing in response to your email dated on March 28, 2022 demanding revised information regarding my manuscript (No: 1653810). Please find below my point to point answers to the questions you raised.

Reviewer 1

This article focuses on interventional techniques for the treatment of pudendal neuroalgia. While the topic is of interest, I believe the article in its present form has some major issues.

Q1 First, in the abstract, what has been identified as an objective is actually a background and the aim of the study is missing.

Answer: Thank you for your kindly reminder. We have revised objective of the study (page 1 line 12-14 in the revised MS)

Q2 In the introduction, the gap of knowledge and aims are not quite well related and some phrases are confusing.  

Answer: We deeply appreciate your suggestion. We have revised the introduction and state our purpose of this manuscript more clarity. (page 2 line 45-64 in the revised MS)

Q3: One of the most relevant issues is the design of the study. While the authors state that this is a retrospective study, from the methods decription it seems like a prospective one with various follow-ups.

Answer: Thank you for your reminder. We initially designed the study as a prospective study, considering the cases of PN patients were insufficient, and the new technologies appeared in our department of pain medicine. To better evaluate the efficacy and safety of improved ultrasound-guided pulsed radio frequency (PRF)/nerve block (NB), we collected data on all hospitalized patients with pudendal neuralgia in a single center over 10 years. We think it is still a retrospective study base on the ethic principles and statistical analysis needs.

Q4: Additionally, in the data collection, authors state they used the Visual Aanalogue Scales (VAS) scale to evaluate pain intensity, collected telephonically, which to my knowledge is not feasibile. 

Answer: Thank you for your comments. All patients included in the study were hospitalized in the Department of Pain Medicine for more than 1 week. They were assessed the degree of the pain in the ward every day and were very familiar with the test. Most of the patients could tell us the scale of pain just like in the hospital on the phone or WeChat video call. Therefore, although we could not use the VAS rule to evaluate pain level face-to-face, it was thought to be the VAS scale test. (page 3 line 99-100 in the revised MS)

Q5: In the statistical analysis too, it is unclear which timepoint was used as post-treatment score for the primary outcome.

Answer: Thank you for your reminder. We used paired t-tests to compare treatment VAS values and pain severity at different post treatment time points (1d,3d,7d,14d,1M,3M)with those at baseline (pretreatment), respectively, to analyze whether there were significant changes in pain severity pre/post treatment. (page 6 line 164-167 in the revised MS)

Q 6: It is also not clear how were data of patients who early terminated follow-up treated.

Answer: We selected the study population were patients who had follow-up treatment for 3 months, since it's a retrospective observational study. Patients who early terminated follow-up treated were excluded from the study dataset.

Submission Date

10 March 2022

Date of this review

18 Mar 2022 14:23:40

Thank you for your precious comments and advice. Those comments are all valuable and very helpful for revising and improving our paper. We have revised the manuscript accordingly, and our point-by-point responses are presented above. Please feel free to contact us if you have any further concerns.

Thank you for your serious consideration of our work.

Best Regards,

Jian Cui, MD, PHD

Department of Pain Medicine, Southwest Hospital, Army Medical University, Chongqing 400038, China

Reviewer 2 Report

I have some questions about your work:

1)In line 78 you state "symptoms only presented with  tail bone, hip or stomach pain and occasionally itching". This sentence must be clarified, what is the relationship between stomach pain an pudendal neuralgia? What do you mean with occasionally itching? 

2) How did you allocate patients for conventional (blind) or ultrasound guided nerve block and for PRF?Randomly or with other criteria?

3)You should add in the results the percentage of patients with pain relief>50% from baseline to define whether the procedure has been successful or not. This in an important outcome and gives more informations compared to reporting only mean VAS values.

Author Response

Apr. 6, 2022

Dear editor,

We are very grateful to Editors and Reviewers for reviewing the paper so carefully. We have carefully considered the suggestions of yours and make some changes. I am writing in response to your email dated on March 28, 2022 demanding revised information regarding my manuscript (No: 1653810). Please find below my point to point answers to the questions you raised.

Reviewer 2

I have some questions about your work:

Q1) In line 78 you state "symptoms only presented with tail bone, hip or stomach pain and occasionally itching". This sentence must be clarified, what is the relationship between stomach pain an pudendal neuralgia? What do you mean with occasionally itching? 

Answer: 1) Thank you for your kindly reminder. We mistyped hypogastric pain as stomach pain, which has now been corrected. In Nantes Criteria, this type of pain does not correspond to the anatomical territory of the pudendal nerve.

2) We are worried about causing misunderstandings among readers and have removed the “occasionally”. Itching is primarily suggestive of a dermatological lesion (atrophic lichen planus, etc.) rather than a nerve lesion. We revised the sentence to make it clarity. The concept of Itching includes a need to scratch which is not experienced in the context of pudendal neuralgia. (page 2 line 89 in the revised MS)

Q2) How did you allocate patients for conventional (blind) or ultrasound guided nerve block and for PRF? Randomly or with other criteria?

Answer: We are very sorry for making you confusing about this. It was a retrospective study, and we just collected data on all PN patients over the last decade and assigned them to different groups based on their treatment.

Q3): You should add in the results the percentage of patients with pain relief>50% from baseline to define whether the procedure has been successful or not. This in an important outcome and gives more information compared to reporting only mean VAS values.

Answer: Thank you very much for the good suggestion. We add a table to exhibit the pain relief>50% from baseline to post treatment time points (1d,3d,7d,14d,1M,3M). (page 9-10 line 254-270 in the revised MS)

Submission Date

10 March 2022

Date of this review

26 Mar 2022 15:53:11

Thank you for your precious comments and advice. Those comments are all valuable and very helpful for revising and improving our paper. We have revised the manuscript accordingly, and our point-by-point responses are presented above. Please feel free to contact us if you have any further concerns.

Thank you for your serious consideration of our work.

Best Regards,

Jian Cui, MD, PHD

Department of Pain Medicine, Southwest Hospital, Army Medical University, Chongqing 400038, China

Reviewer 3 Report

I have peer reviewed the manuscript titled: “The efficacy of an ultrasound-guided improved puncture path technique of nerve block/pulsed radiofrequency for pudendal neuralgia: a retrospective study”. The authors described new, innovative ultrasound-guided path technique of pulsed radiofrequency (PRF)/nerve block (NB) for patients with pudendal neuralgia.

In my opinion, the manuscript is written well and describes an interesting from practical point of view topic. In order to improve its practical value, the figures need to be corrected. To be more readable they must be larger. Figure 2 is intended to show the optimal positioning of the ultrasound transducer during blockade / PRF, however the blue drapes make orientation very difficult. Figure 3 with ultrasound scans also should be larger, more readable and should contain the precise identification of anatomical structures, which should be signed with arrows.

Due to the practical importance of the topic discussed in the paper, in my opinion this manuscript should be published in Brain Sciences, but it is necessary to correct the figures as suggested above.

Author Response

Apr. 6, 2022

Dear editor,

We are very grateful to Editors and Reviewers for reviewing the paper so carefully. We have carefully considered the suggestions of yours and make some changes. I am writing in response to your email dated on March 28, 2022 demanding revised information regarding my manuscript (No: 1653810). Please find below my point to point answers to the questions you raised.

Reviewer 3

I have peer reviewed the manuscript titled: “The efficacy of an ultrasound-guided improved puncture path technique of nerve block/pulsed radiofrequency for pudendal neuralgia: a retrospective study”. The authors described new, innovative ultrasound-guided path technique of pulsed radiofrequency (PRF)/nerve block (NB) for patients with pudendal neuralgia.

In my opinion, the manuscript is written well and describes an interesting from practical point of view topic.

Q1: In order to improve its practical value, the figures need to be corrected. To be more readable they must be larger. Figure 2 is intended to show the optimal positioning of the ultrasound transducer during blockade / PRF, however the blue drapes make orientation very difficult. Figure 3 with ultrasound scans also should be larger, more readable and should contain the precise identification of anatomical structures, which should be signed with arrows.

Answer: Thanks a lot for your kindly and professional advice. We modified Figure 2 and Figure 3 to make it more accurately and clarity. (page 4-5 line 123-124, 129-130 in the revised MS)

Submission Date

10 March 2022

Date of this review

27 Mar 2022 23:48:27

Thank you for your precious comments and advice. Those comments are all valuable and very helpful for revising and improving our paper. We have revised the manuscript accordingly, and our point-by-point responses are presented above. Please feel free to contact us if you have any further concerns.

Thank you for your serious consideration of our work.

Best Regards,

Jian Cui, MD, PHD

Department of Pain Medicine, Southwest Hospital, Army Medical University, Chongqing 400038, China

Reviewer 4 Report

Paper is well written and represents an interesting topic. I have some minor concerns to improve:

  • "The exclusion criteria were: 1) tumors of the pelvic or/and sacral channel.. (MRI)" So all patients performed a lumbar MRI bebore nerve block radiofrequency, please state this in the text.
  • "In recent years, ultrasound-guided pudendal nerve block and/or PRF techniques have been used because of their accuracy, visualization, and lack of radiation" But also new technology as virtual reality in the operating room. See refs: doi: 10.1186/s13063-019-3922-2 -- doi: 10.3390/ijerph19031719 .
  • Paper has some limitations, as for example the small sample and other thing. Please add a limitation section at the end of the discussion.
  • Did authors have any complications after nerve block/pulsed radiofrequency procedures? If no, please state "No complications were reported".
  • Lines 298-309 no references appear. Please add.
  • Statistical analysis appears correct

Author Response

Apr. 6, 2022

Dear editor,

We are very grateful to Editors and Reviewers for reviewing the paper so carefully. We have carefully considered the suggestions of yours and make some changes. I am writing in response to your email dated on March 28, 2022 demanding revised information regarding my manuscript (No: 1653810). Please find below my point to point answers to the questions you raised.

Reviewer 4

Paper is well written and represents an interesting topic. I have some minor concerns to improve:

Q1: "The exclusion criteria were: 1) tumors of the pelvic or/and sacral channel.. (MRI)" So all patients performed a lumbar MRI bebore nerve block radiofrequency, please state this in the text.

Answer: Thank you for your kindly suggestion. we add a specific criterion in inclusion protocol to state this. (page 2 line 84-85 in the revised MS)

Q2: "In recent years, ultrasound-guided pudendal nerve block and/or PRF techniques have been used because of their accuracy, visualization, and lack of radiation" But also new technology as virtual reality in the operating room. See refs: doi: 10.1186/s13063-019-3922-2 -- doi: 10.3390/ijerph19031719 .

Answer: Thank you for the kindly reminder. We recognize the importance of virtual reality (VR) in improving the technique in operating room and other field in the nearly future. However, it has not yet been widely used in clinical practice. We note that the first literature provided evidence of the application of VR in patients with a relaxing virtual environment to alleviate intraoperative anxiety during surgery and the second review illustrated that the application prospects and potential use of VR technology in neurosurgery. We are fully confident that VR will help us to make the treatment of pain more accurate, effective, and simpler. We will also try and strengthen the learning and application of VR in the future.

To avoid ambiguity, we add ‘widely’ in front of ‘used’ in this sentence as shown in the text. (page 2 line 48 in the revised MS)

Q3: Paper has some limitations, as for example the small sample and other thing. Please add a limitation section at the end of the discussion.

Answer: Thank you for your kindly suggestion. We discuss the limitations of the study and further studies are looked forward to. (page 12 line 367-369 in the revised MS)

Q3: Did authors have any complications after nerve block/pulsed radiofrequency procedures? If no, please state "No complications were reported".

Answer: Thank you for this reminder. We add this statement to the end of Results. (page 10 line 272-273 in the revised MS)

Q4: Lines 298-309 no references appear. Please add.

Answer: Thank you for your suggestion. We have added two related literatures on sacral perineural cyst in this paragraph. (page 12 line 348-354 in the revised MS)

Q5: Statistical analysis appears correct.

Answer: Thank you very much for your comment.

Submission Date

10 March 2022

Date of this review

21 Mar 2022 16:54:16

Thank you for your precious comments and advice. Those comments are all valuable and very helpful for revising and improving our paper. We have revised the manuscript accordingly, and our point-by-point responses are presented above. Please feel free to contact us if you have any further concerns.

Thank you for your serious consideration of our work.

Best Regards,

Jian Cui, MD, PHD

Department of Pain Medicine, Southwest Hospital, Army Medical University, Chongqing 400038, China

Round 2

Reviewer 2 Report

Thank you for your work which has significantly improved the paper